# Comparative Analysis of Bone Regeneration According to Particle Type and Barrier Membrane for Octacalcium Phosphate Grafted into Rabbit Calvarial Defects

**DOI:** 10.3390/bioengineering11030215

**Published:** 2024-02-24

**Authors:** Se-Wook Pyo, Jeong-Won Paik, Da-Na Lee, Young-Wook Seo, Jin-Young Park, Sunjai Kim, Seong-Ho Choi

**Affiliations:** 1Department of Prosthodontics, Gangnam Severance Dental Hospital, Yonsei University College of Dentistry, Seoul 06273, Republic of Korea; dentipyo@yuhs.ac (S.-W.P.); sunjai@yuhs.ac (S.K.); 2Department of Periodontology, Research Institute for Periodontal Regeneration, Yonsei University College of Dentistry, Seoul 03722, Republic of Korea; jpaik@yuhs.ac (J.-W.P.); danalee@yuhs.ac (D.-N.L.); ahkupechia07@gmail.com (Y.-W.S.); jypark87@yuhs.ac (J.-Y.P.)

**Keywords:** bone regeneration, octacalcium phosphate, concentration, collagen membrane

## Abstract

This animal study was aimed to evaluate the efficacy of new bone formation and volume maintenance according to the particle type and the collagen membrane function for grafted octacalcium phosphate (OCP) in rabbit calvarial defects. The synthetic bone substitutes were prepared in powder form with 90% OCP and granular form with 76% OCP, respectively. The calvarial defects were divided into four groups according to the particle type and the membrane application. All specimens were acquired 2 weeks (n = 5) and 8 weeks (n = 5) after surgery. According to the micro-CT results, the new bone volume increased at 2 weeks in the 76% OCP groups compared to the 90% OCP groups, and the bone volume ratio was significantly lower in the 90% OCP group after 2 weeks. The histomorphometric analysis results indicated that the new bone area and its ratio in all experimental groups were increased at 8 weeks except for the group with 90% OCP without a membrane. Furthermore, the residual bone graft area and its ratio in the 90% OCP groups were decreased at 8 weeks. In conclusion, all types of OCP could be applied as biocompatible bone graft materials regardless of its density and membrane application. Neither the OCP concentration nor the membrane application had a significant effect on new bone formation in the defect area, but the higher the OCP concentration, the less graft volume maintenance was needed.

## 1. Introduction

Bone grafts are divided into autograft, allograft, xenograft, and alloplast according to the origin of their material, and each is used individually or in combination depending on the purpose in the field of bone regeneration [1]. Many studies related to synthetic bone graft materials, which have various advantages such as mass production, controllability of properties, and no risk of cross-infection, are continuously being conducted, and new materials are being developed [2]. It is essential for bone substitute materials to provide a variety of shapes and sizes with mechanical strength and biocompatibility to facilitate their use in the regeneration of bone defects [3]. In particular, several studies have been reported on hydroxyapatite (HA) and β-tricalcium phosphate (β-TCP), which are calcium phosphate-based biomaterials that are typically used as synthetic bone materials [4]. Sintered HA bone is used as a scaffold material because it does not dissolve and maintains its shape in a long-term bone defect. Compared with HA, β-TCP is a resorbable bone substitute with inherent solubility in physiological environments. For this reason, biphasic calcium phosphate (BCP), which consists of HA and β-TCP phases in various ratios, is mostly capable of controlling the resorption rate [5]. MBCP+^TM^ (Biomatlante SAS, 88 Édouard Belin, Vigneux de Bretagne, France) is a synthetic bone graft substitute under the BCP category, featuring a 20:80 ratio of HA to β-TCP. It possesses a micro- and macro-porous structure that closely mirrors the natural architecture of human bone [6,7].

Octacalcium phosphate (OCP; Ca_8_H_2_[PO_4_]_6_·5H_2_O), recently developed as a new synthetic bone substitute, is a direct precursor of biological apatite [8]. The OCP shows outstanding osteoconductive and osteoinductive properties owing to its unique physicochemical properties. The OCP crystal has a water layer between two apatite layers, so that the water layer is removed from the OCP in a physiological environment, and the two apatite layers are combined to form HA crystals [9]. The OCP could be irreversibly converted into sustainable biological apatite under physiological conditions, which would be biodegradable at a bone defect site with a neutral pH [10,11]. Therefore, OCP has properties as a biodegradable substance while also facilitating bone development through osteoblast activity. There are several reports that biodegradation through direct resorption of osteoclast-like cells was observed at the OCP-grafted site in animal bone defect models [12,13,14]. Despite this OCP biodegradability, most granules remain intact because the degradation rate of OCP depends on granule size [15]. The resorption of bone substitutes is influenced by several factors. These factors include the size of particles, their porosity, chemical and crystallographic properties (composition, Ca/P ratio, phase, and crystallinity), and compatibility with body fluids, as indicated by pH [16]. Resorption of substitutes is usually faster with smaller particle size, higher porosity, lower crystallinity, and a higher non-stoichiometric ratio [17]. The rapid rate of resorption of OCP is an essential consideration for clinicians.

In addition, a key principle of guided bone regeneration (GBR) involves creating a secluded area for new bone formation by obstructing the encroachment of soft tissues [18]. In order to provide space for the stability of various bone substitutes, it is necessary to use a barrier membrane that maintains structural integrity during the periods of cell proliferation and maturation inside the wound. Particularly for resorbable membranes, it is ideal for the rate of biodegradation to match the rate of new tissue formation without leaving any residual material [19]. Collagen membranes can be explained by their well-established scientific background and extensively validated clinical use [20]. Nevertheless, to our knowledge, there has been no study on a GBR with different OCP concentrations or collagen membrane application.

Therefore, the purpose of this animal study was to investigate the efficacy of new bone formation and volume maintenance according to the OCP concentration and the collagen membrane function in the GBR of the rabbit calvarium.

## 2. Materials and Methods

### 2.1. Material Preparation

Two types of synthetic bone substitutes, in powder and granular form respectively, were prepared. The relative content of OCP was found to be 90% in powder and 76% in granules. OCP powder was synthesized using a meticulously controlled process. The procedure involved dispersing dicalcium phosphate dihydrate (DCPD; Junsei Co., Ltd., Tokyo, Japan), a carefully selected precursor, into deionized water to create a finely dispersed suspension with a concentration of 12 g/L. Since the pH of the calcium phosphate solution increases as the basic calcium phosphate hydrate is mixed with deionized water, an acidic solution was added when the pH increased to adjust the pH of the calcium phosphate solution to between 5 and 6 in the initial stage. Thereafter, the calcium phosphate solution was heated to a temperature of 90° C or lower, and as a result, evaporation of the solvent from the calcium phosphate solution was prevented, thereby suppressing a decrease in the stability of the calcium phosphate solution. In addition, by adding a heterogeneous nucleus containing at least one form of OCP and DCPD to the heated calcium phosphate solution, OCP crystals were able to grow at a rapid rate centered on the heterogeneous nucleus; that is, the crystal nucleus. As acidic OCP crystallizes in the calcium phosphate solution, the pH of the calcium phosphate solution decreases to less than 5. Even in the later stages, a basic solution such as ammonia water is added to maintain the pH of the calcium phosphate solution. And then the solution was passed through filter paper. Since the residue left on the filter paper included OCP crystals precipitated from the calcium phosphate solution, it was washed with a solvent to remove impurities other than the OCP crystals contained in the residue. The washed residue was then dried at a temperature of 70 °C to 90 °C, preferably 80 °C, for about 24 h, and as a result, OCP crystals with high purity were obtained. Two sizes of crystals with non-uniform size can be obtained from the OCP prepared by this method; one crystal with a length of 10–15 μm and a width of 2–3 μm, and a second crystal with a length of 1–10 μm and a width of 0.5–3 μm. This method of mechanically grinding a mixture of relatively small and large OCP crystals allows the OCP crystals to be sized to the desired size. In this study, a meticulous particle formation process was employed within a controlled environment, utilizing advanced granulation techniques. OCP powder was prepared by hydrating DCPD as the raw material with water in a pH-controlled system using acetic acid. The OCP granule was also prepared by cementation of a mixture of α-tricalcium phosphate (α-TCP) and sodium phosphate. The Ca/P ratio of the paste that was poured into the rectangular mold was 1.00. The dried sample was ground and sieved to a size of 500–850 μm [21]. Throughout the entire manufacturing process, which adhered to a biomimetic approach, the temperature was carefully maintained below 100 °C to preserve the intrinsic properties of the material. Moreover, the OCP material employed in this study underwent rigorous gamma irradiation sterilization (25–40 kGy) to ensure the highest standards of safety and quality. The OCP material was inserted into sterilized conical tubes (1.5 mL; 509-GRD-SC, Quality Scientific Plastics, San Diego, CA, USA) and then packaged in Fisherbrand instant-sealing sterile pouches (9 cm × 13 cm; Fisher Scientific, Pittsburgh, PA, USA). Then, the packaged OCPs were sterilized by gamma irradiation into a ready-to-use form. The OCP granules mounted by a sample holder or the OCP powders ground using an agate mortar and pestle were characterized using an X-ray diffractometer (XRD, X’pert MPD-PRO; Panalytical, Almelo, Netherlands).

### 2.2. Material Characterization

Phase analysis was conducted using X-ray diffraction (XRD; AERIS Malvern Panalytical Co., Ltd., Malvern, UK) with a Cu-Kα radiation source at a scan speed of 1.0°/min and a step size of 0.022°. The reference database was based on the inorganic crystal structure database (ICSD) [22]. The references for each substance are OCP (04-016-3473), HA (00-009-0432), β-TCP (00-009-0169), DCPD (00-009-0077). Surface morphology was observed using field-emission scanning electron microscopy (FE-SEM, S-4700 Hitachi, Tokyo, Japan).

### 2.3. Animals and Surgical Protocol

This study used ten male New Zealand white rabbits, each weighing between 2.8 and 3.2 kg. They were individually housed in separate cages under controlled laboratory conditions and fed a standard diet. Every step, from the selection and care of the animals to their preparation and the surgical procedures, was conducted in accordance with a protocol sanctioned by the Institutional Animal Care and Use Committee, Yonsei Medical Center, Seoul, Korea (approval number 2021-0237). All the protocols followed the ARRIVE (Animal Research: Reporting of In vivo Experiments) guidelines for the study design [23,24,25].

### 2.4. Study Design

Four circular defects measuring 8 mm in diameter were created in the calvarium of each rabbit using the same method as in previous studies [26,27]. Two different types of synthetic bone substitutes, in powder and granular form, respectively, were applied to the calvarial defects. In the groups using a barrier membrane, a resorbable collagen membrane (Bio-Gide^®^, Geistlich Pharma AG, Wolhusen, Switzerland; LOT number 82100857) was cut into the size of 10 mm × 10 mm and placed over each defect. As a result, the defects on the calvarium were randomly assigned to one of the following four experimental groups; (1) 90m: 90% OCP with a membrane; (2) 90n: 90% OCP without a membrane; (3) 76m: 76% OCP with a membrane; and (4) 76n: 76% OCP without a membrane. The first five rabbits were sacrificed at 2 weeks (n = 5) and the other five at 8 weeks (n = 5) after surgical procedure. The sample size was calculated based on a previous study (Figure 1).

### 2.5. Surgical Protocol

The rabbits were anesthetized with intramuscular injections of zoletil (15 mg/kg) and rompun (5 mg/kg) maintained with inhalations of isoflurane (2–2.5%). The surgical sites were shaven and then draped with alcohol and povidone iodine, followed by local anesthesia with 2% lidocaine. An incision was made along the sagittal midline from the frontal bone to the occipital bone. A full-thickness flap was elevated to expose the cranial bone. Four circular defects with a diameter of 8 mm were created using a trephine bur of corresponding size under copious saline irrigation, and randomly allocated to the four study groups (Figure 2). After surgery, the animals were managed by administering antibiotics subcutaneously (Baytril, Bayer, Leverkusen, Germany) and were closely monitored clinically during the entire period of healing. The animals were sacrificed at either 2 or 8 weeks postoperatively.

### 2.6. Evaluation

−Clinical observations: Animals were meticulously monitored for a duration of 2 to 8 weeks post-surgery, assessing for potential complications such as inflammation, allergic reactions, postoperative bleeding, and infections around the surgical site.−Micro-computed tomography (micro-CT) analysis: After the animals were sacrificed, the surgical defects and adjacent tissues were removed in a single piece. These samples were rinsed in sterile saline and preserved in 10% buffered formaldehyde solution for 10 days. After rinsing in water, all specimens were scanned using a micro-CT scanner (SkyScan1173;Bruker-CT, Kartuizersweg 3B 2550 Kontich, Belgium). The samples were mounted on a jig using parafilm for precise micro-CT analysis, during which 800 images were captured at a setting of 130 kV tube voltage, 60 µA current, and through a 1.0 mm aluminum filter. Image reconstruction was performed using NRecon software (Ver. 1.7.0.4, Bruker, Kontich, Belgium), with alignment of the cross-sectional images for each specimen achieved through DataViewer software (Ver. 1.5.1.2, Bruker, Kontich, Belgium), and quantitative analysis conducted using CTAn software (Ver. 1.17.7.2, Bruker, Kontich, Belgium). For data analysis, the defect was set as the region of interest, and the threshold was set to 45–90 to analyze the following parameters:
Total tissue volume (TV; mm^3^): total augmented volume of the defects.New bone volume (BV; mm^3^): volumetric measurements of the new bone within the defects.Bone volume ratio (BV/TV; %): new bone volume to total augmented volume.−Histologic and histomophometric analysis: Following micro-CT scanning, the specimens underwent a 14-day decalcification process in 5% formic acid and were subsequently embedded in paraffin. Serial 5 μm thick sections were obtained from the central region of each calvarial defect. The central sections from each block were subjected to hematoxylin and eosin (H&E) and Masson trichrome staining for histological and histometric analyses. One blinded examiner conducted examinations using a microscope (DM LB, Leica Microsystems, Wetzlar, Germany) equipped with a camera (DC300F, Leica Microsystems). The slide images were saved as digital files, and computer-aided histometric measurements were performed using an automated image analysis system (Image-Pro Plus; Media Cybernetics, Silver Spring, MD, USA). The assessment of bone healing and regeneration involved measuring the following parameters in each histological section corresponding to the defect areas:
Total augmented area (TA; mm^2^): Total sum of the area of new bone, residual particles, connective tissue, adipose tissue, blood vessels within the defect area.New bone area (NB; mm^2^) and its ratio (%NB): Area of the newly formed bone within the defect, and ratio of NB to TA.Residual bone graft area (RG; mm^2^) and its ratio (%RG): Area of the residual material within the defect, and ratio of RG to TA.

### 2.7. Statistical Analysis

A power calculation was conducted using a power analysis program (G*Power 3.1 software, University of Kiel, Kiel, Germany), focusing on new bone area (NB) as the primary outcome. Following the findings of Sohn et al., a sample size of 10 subjects was recommended (assumed mean and standard deviation: 2 weeks, 1.88 ± 0.67 mm^2^; 8 weeks, 3.19 ± 0.27 mm^2^; power: 0.80; α < 0.05) [28]. The descriptive data from radiographic and histometric results were expressed as median values, and difference between the maximum and minimum values. Data were analyzed using SAS version 9.4 (SAS Institute, Cary, NC, USA). The significance between four different groups at each time point were determined by the Kruskal–Wallis test followed by the Mann–Whiney test. The Mann–Whiney test was also used for comparisons between the two healing periods within the same experimental group. Statistical significance was determined at the *p* < 0.05 level.

## 3. Results

### 3.1. Phase Analysis Using X-ray Diffration

The OCP synthetic powder and granules, produced through biomimetic process, demonstrated the absence of residual DCPD. Notably, X-ray diffraction analysis conducted within the 2θ range of 3–60° revealed the distinct presence of an OCP diffraction peak (100), prominently observed at approximately 4–5° (Figure 3A). In distinction to the OCP synthesized immediately after production, the OCP derived through the granulation process demonstrated the emergence of minute quantities of amorphous calcium phosphate (ACP) ceramics under humid conditions. However, it can be observed that the inherent XRD pattern of OCP is well-preserved, indicating its structural stability throughout the granulation process. Phase analysis using XRD showed that OCP is the main constituent of both materials, regardless of size.

In order to investigate the surface properties of the bone graft material, scanning electron microscopy (SEM) analysis was employed (Figure 3B). The OCP particles exhibited a delicate ribbon-like structure, displaying a wide range of sizes from a few micrometers to several tens of micrometers. The agglomeration of these ribbon-shaped OCP particles was observed, resulting in the formation of intricate micro-pores. In contrast, the granulated form exhibited comparatively smaller particles, likely attributable to additional processes.

### 3.2. Clinical Findings

No significant postoperative complications, such as excessive bleeding or swelling, were observed. During the healing period, no specific inflammation, complications, or abnormal findings were observed in any animal. Upon sacrifice and sample collection, it was verified that the grafted materials remained intact within the defects.

### 3.3. Micro-CT Analysis

The new bone tissue tended to grow toward the center of the defect region. Although all groups showed new bone formation in the bone defects, none of the defects were completely closed after 2 or 8 weeks (Figure 4).

The TV results were similar in all groups without significant difference, and decreased at 8 weeks compared to 2 weeks with a statistically significant difference only in the 76n group. The BV results were significantly greater in the 76% groups than in the 90% groups at 2 weeks, and the difference between the groups decreased at 8 weeks. The BV/TV results also tended to be greater in the 76% groups than in the 90% groups at 2 weeks, but they showed similar results in all groups at 8 weeks. As a result, significant increases were observed in the BV/TV results between 2 and 8 weeks in the 90% groups, but no differences were observed in the 76% OCP groups. The results of the micro-CT analysis are summarized in Figure 5 and Table 1.

### 3.4. Histological Analysis

The defect spaces were occupied by the underlying brain tissues, and either the overlying periosteum or collagen membrane collapsed to fill the vacant space. The grafted materials were surrounded by the connective tissue of the extracellular matrix (ECM) comprising a loose collagen network. After 2 weeks, the infiltration of chronic inflammatory cells and proliferation of blood vessels were observed. Minimal new bone formation was observed around the grafted particles in the margins adjacent to the defect. The 76% OCP groups showed a stable volumetric change in total tissue volume, and the formation of new bone around the defect margin, and grafted particles were observed due to the support of relatively large particles. In contrast, in the groups with 90% OCP, as a result of being strongly pressed by soft tissue and dura mater in the center of the defect, a thin, flat, concave, disc-shaped tissue was observed. In the 8-weeks group, larger quantities of new bone and osteoblasts were noted, predominantly originating from the periphery of the defects. New bone was observed not only growing from the marginal area but also around the centrally located remaining particles. The collagen membrane maintained a uniform shape, but was absorbed compared to the findings at 2 weeks, and the amount of chronic inflammatory cells around the membrane also decreased. ECM exhibited increased density compared to the appearance observed after 2 weeks, resulting in almost complete closure along the outer surface of the defect (Figure 6).

### 3.5. Histomorphometric Analysis

Except for a significant difference between the 90m (11.755 mm^2^) and 76n (16.720 mm^2^) groups at 8 weeks, most of the TA results were measured consistently between the experimental groups as well as between the findings at 2 and 8 weeks. The overall median values representing TA in all groups were 14.890 mm^2^ for 2 weeks and 14.315 mm^2^ for 8 weeks. Statistically significant increases in NB and %NB were observed at 8 weeks compared to 2 weeks in all groups except for the 90n group. In all groups, the overall median values of NB increased from 0.975 mm^2^ for 2 weeks to 2.290 mm^2^ for 8 weeks, and the values of %NB, the ratio of NB to TA, increased from 5.985% for 2 weeks to 16.128% for 8 weeks. However, there was no significant difference between the experimental groups within the same period. On the other hand, RBG and %RBG showed a tendency to decrease more at 8 weeks than at 2 weeks, including statistical significance only in the 90n group. The overall median values of RBG and %RBG decreased from 2.405 mm^2^ and 15.3% for 2 weeks to 0.800 mm^2^ and 6.034% for 8 weeks, respectively. At 8 weeks, the RBG and %RBG of the 76m group (3.580 mm^2^/24.323%) were higher than those of the 90% OCP groups with a significant difference. The 76n group (3.615 mm^2^/21.968%) also showed higher RBG and %RBG than the 90% OCP groups, but there was no statistical significance due to large differences between animals. The results from the histomorphometrical analysis are summarized in Figure 7 and Table 2.

## 4. Discussion

OCP has been proven to be effective in new bone formation due to its high bone regeneration ability and rapid bioabsorption compared to xenogeneic bone or other synthetic bone [2]. Earlier investigations have been conducted to assess the efficacy of innovative biomaterials derived from amorphous calcium phosphate (ACP), octacalcium phosphate (OCP), and hydroxyapatite (HA) in addressing critical size defects in the rabbit skull. Therefore, this study did not establish separate positive and negative controls [29].

According to the solubility isotherm, calcium phosphate materials demonstrate a solubility sequence in the order of HA < TCP < OCP < DCPD (dicalcium phosphate dehydrate) at pH 7.4. Although a high osteogenesis capacity is more important than solubility, OCP, which has higher solubility than TCP, can be considered advantageous for inducing new bone tissue [30]. In other words, OCP causes a rapid bone regeneration compared to TCP because it has high biodegradability (rate of disappearance) through cellular phagocytosis in vivo and at the same time has an excellent bone formation ability. The basic theory for using BCP as a commonly used synthetic bone substitute is that new bone formation and resorption can be regulated by the HA:TCP ratio [31]. For BCP, the degree of solubility depends on the β-TCP/HA ratio, with higher ratios increasing the solubility [32,33]. In the case of MBCP, it is suggested to mix β -TCP and HA in a ratio of 80:20% to form hard particles that are easy to handle for application to the defect [34].

OCP has been difficult to develop as a synthetic bone substitute because its original crystal structure could be changed during the sintering process, making it brittle and fragile, and unable to be formed into larger solid masses [35]. These limitations have led most of studies on the commercialization of OCP-based bone substitutes to focus on coating OCP onto DBBM or combining it with collagen [36,37]. Previous investigations on OCP encompassed studies involving OCP granules and composites with biodegradable polymers like collagen, gelatin, hyaluronic acid, alginate, polycaprolactone (PCL), chitosan, silk fibroin, and polylactic-co-glycolic acid (PLGA) [38]. Recently, a novel synthesis method, eliminating the need for sintering, has been devised to enable the large-scale production of high-purity OCP with enhanced physical properties [3]. However, studies on the difference in osteogenesis ability depending on the concentration of OCP are still difficult to find as far as we know.

The usefulness of 8 mm defect formation in rabbit calvaria for new bone formation has been fully validated in previous studies [28]. In this study, four defects with a diameter of 8 mm were formed in rabbit calvaria, and the new bone formation ability according to the difference in OCP concentration and whether or not the collagen membrane was applied was compared after 2 and 8 weeks, respectively. The two concentrations of commercial OCP synthetic bone substitutes used in this study were prepared by mixing OCP and HA in weight ratios of 90:10 and 76:24, respectively. In the micro-CT results, significantly greater amounts of BV and BV/TV in the 76% groups than in the 90% groups at 2 weeks demonstrated that osteogenesis was rapidly induced in the OCP group containing more HA as a scaffold. Although significantly increased BV and BV/TV were observed at 76% OCP at week 2, this result does not necessarily mean rapid new bone formation, which means that the grafted space is relatively sufficiently maintained. According to the histomorphometric results, new bone formation indices such as NB and %NB were greater at 8 weeks compared to 2 weeks, but there was no significant difference between the experimental groups. However, the reduced differences in RBG and %RBG between the 2- and 8-week results were significantly greater in the 90% groups. In the 76% groups, the graft material was not absorbed to a significant extent over time and was mostly maintained. As a result, the indices related to the RBG showed significantly greater results in the 76% groups than in the 90% groups at 8 weeks. A thin, flat, concave, disc-shaped tissue observed in histological findings for the 90% groups at 2 weeks also showed that high concentrations of OCP were related to rapid biodegradation.

A resorbable collagen membrane was also used in this study, considering the characteristics of synthetic bone graft materials that lack rigidity in terms of maintaining the defect space. As a result, it was observed that the two types of OCP maintained the graft material stably, without any histologically specific complications or abnormal findings, regardless of whether the collagen membrane was used or not. An ideal bone substitute material should stabilize the space of the defect during the period of new bone formation, which is closely related to the concentration of the bone graft material [39]. Regarding the bone substitute material used in this study, 76% OCP containing 24% HA could be applied as granules of relatively large particles, while 90% OCP containing 10% HA was difficult to apply as a slurry type. Although there was no statistical significance, all of the indices related to NB and RBG at 2 weeks showed greater values in the membrane-application groups (90m, 76m) than in the non-application groups (90n, 76n) This fact suggests that the membrane is effective in keeping the initial OCP volume stable. Therefore, despite the faster bone regeneration effect in high-concentration OCP, it is difficult to continuously maintain the augmented volume, so the use of a collagen membrane for long-term stability is effective.

In all experimental groups in this study, a similar degree of new bone formation was observed at 8 weeks, but it was confirmed that there were differences in the amount of remaining graft material and the rate of new bone formation depending on the concentration of OCP. In general, bone regeneration in rabbits progresses at about three times the speed of humans [40]. Therefore, 2 weeks of observation in rabbits corresponds to 1.5 months in humans and 8 weeks of observation in rabbits corresponds to 6 months in humans. In cases where OCP is applied for GBR, it is recommended to adjust the OCP concentration ensuring that the resorption rate of the graft material is similar to the rate of new bone formation. Compared to previous studies using OCP, this study has value as an in vivo experiment that directly compares the results depending on the application method of OCP bone substitute and whether a collagen membrane is applied and determines a clinically efficient method.

## 5. Conclusions

In conclusion, OCP synthetic bone substitutes with two different compositions all showed excellent osteoconduction efficacy, but there were differences in their space-maintaining capabilities depending on the OCP concentration. Although it did not have a significant effect on new bone formation, the use of collagen membranes could be effective when using a high concentration of OCP, which lacks the capability of maintaining space. Further research will be needed to determine the most suitable concentration for OCP to exhibit osteoconductive efficacy while maintaining the defect space.

## Figures and Tables

**Figure 1 bioengineering-11-00215-f001:**
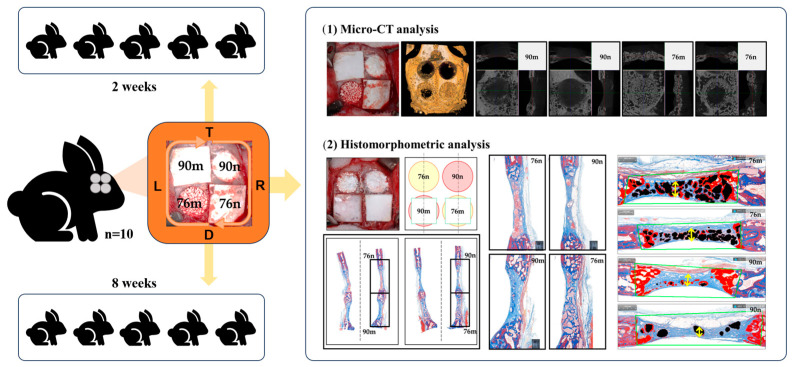
Schematic diagram of this study.

**Figure 2 bioengineering-11-00215-f002:**
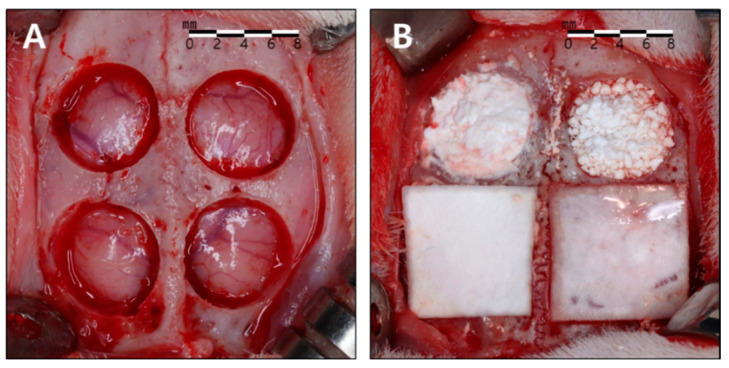
Surgical procedure. (**A**) Four circular defects measuring 8 mm in diameter were created using a trephine bur. (**B**) The defects were allocated to the four study groups. Clockwise from bottom to left: 90% OCP with a membrane (90m), 90% OCP without a membrane (90n), 76% OCP without a membrane (76n), 76% OCP with a membrane (76m).

**Figure 3 bioengineering-11-00215-f003:**
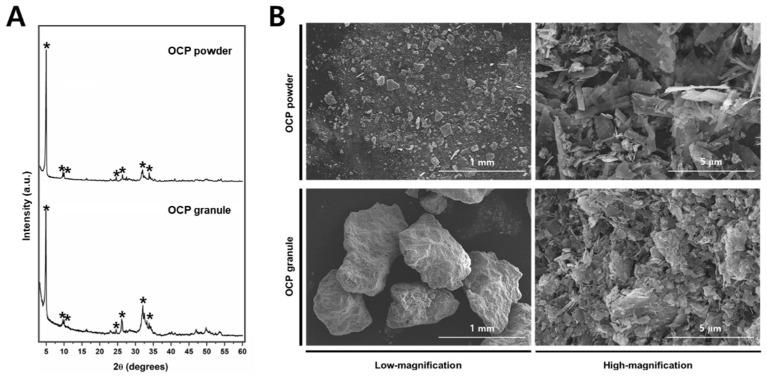
(**A**) The XRD spectra of the OCP powder (**top**) and granules (**bottom**). The OCP peaks are indicated by asterisks (*). (**B**) Representative high- and low-magnification FE-SEM images of the OCP power and granule.

**Figure 4 bioengineering-11-00215-f004:**
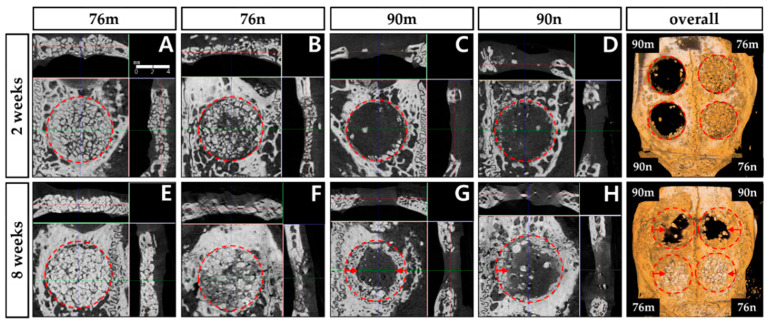
Micro-CT images obtained 2 and 8 weeks after surgery. New bone formation occurred mainly around the periphery of defect, and significantly increased at 8 weeks compared to 2 weeks in all groups. The red circle represents the border of the defect, and the red arrow indicates the direction of presumed new bone formation. (**A**,**E**) 76% OCP with a membrane group; (**B**,**F**) 76% OCP without a membrane group; (**C**,**G**) 90% OCP with a membrane group; (**D**,**H**) 90% OCP without a membrane group; (**right top and down**) three-dimensional reconstruction images.

**Figure 5 bioengineering-11-00215-f005:**
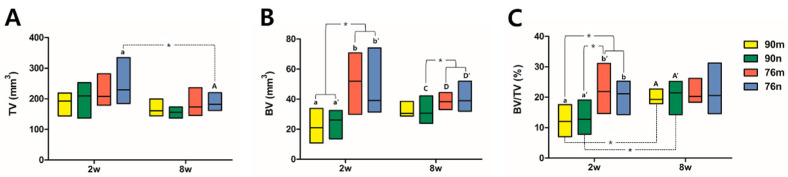
Distribution of experimental groups and statistical significance within groups according to micro-CT analysis results. Small and capital letters of the same alphabet indicate significant differences between 2-week and 8-week results compared in the same experimental group. Even though there is no significant difference, if a distinction is needed, a prime mark is used next to the letter. The asterisk * indicates statistical significance. (**A**) Total tissue volume, (**B**) new bone volume, (**C**) bone volume ration.

**Figure 6 bioengineering-11-00215-f006:**
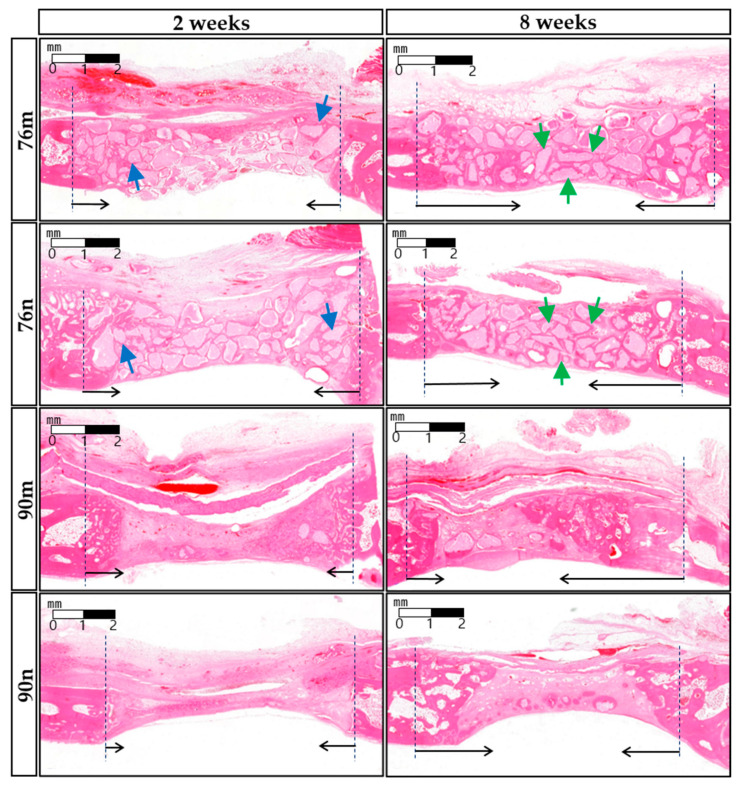
Representative histological images of each experimental groups after 2 or 8 weeks. (HE stain, bar = 1 mm) At 2 weeks, the new bone formation was observed around the grafted particles in the adjacent margin of the defect (blue arrows). At 8 weeks, however, new bone was observed not only growing from the marginal area but also around the centrally located remaining particles (green arrows). The border of the defect is indicated by black dotted lines, and the direction of new bone formation is indicated by black arrows.

**Figure 7 bioengineering-11-00215-f007:**
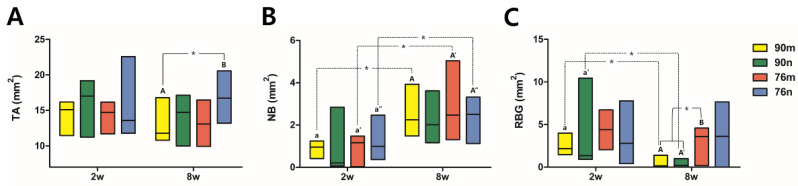
Distribution of experimental groups and statistical significance within groups according to histomorphometric analysis results. Small and capital letters of the same alphabet indicate significant differences between 2-week and 8-week results compared in the same experimental group. Even though there is no significant difference, if a distinction is needed, a prime mark is used next to the letter. The asterisk indicates statistical significance. (**A**) Total augmented area, (**B**) new bone area, (**C**) residual bone graft area.

**Table 1 bioengineering-11-00215-t001:** Median with minimum and maximum values of all groups measured by micro-CT analysis.

Week	Group		TV	BV		BV/TV
2 weeks(n = 5)	90m		192.563 (144.331–219.075)	21.038 (10.813–33.831) ^a^	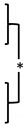	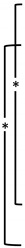	12.004 (7.051–17.569) ^a^	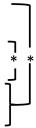
90n		209.308 (136.747–252.565)	26.084 (13.647–32.505) ^a′^	12.787 (7.859–19.074) ^a′^
76m		208.007 (179.307–281.868)	52.035 (29.927–70.808) ^b^	21.899 (14.640–31.131) ^b′^
76n	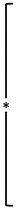	229.541 (184.888–334.893) ^a^	39.112 (31.501–74.109) ^b′^	21.154 (14.334–25.276) ^b^
Overall	208.657 (136.747–334.893)	32.401 (10.813–74.109)		15.688 (7.051–31.131)	
8 weeks(n = 5)	90m	161.167 (144.422–199.540)	30.565 (28.767–38.564)		19.327 (17.849–22.715) ^A^	
90n	155.934 (136.747–173.027)	30.685 (23.884–42.150) ^C^	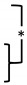	21.449 (14.324–25.173) ^A′^	
76m	173.027 (145.817–236.518)	38.200 (33.107–44.354) ^D^		20.279 (18.388–26.197)	
76n	181.749 (161.864–219.773) ^A^	39.009 (32.037–52.048) ^D′^		20.592 (14.577–31.214)	
Overall		166.051 (136.747–236.518)	34.318 (23.884–52.048)			20.533 (14.324–31.214)	

TV: Total tissue volume, BV: New bone volume, BV/TV: ratio of BV to TV. Significant differences (*p* < 0.05) between the four experimental groups are indicated by different small letters at 2 weeks and by different capital letters at 8 weeks. Small and capital letters of the same alphabet indicate significant differences between 2-week and 8-week results compared in the same experimental group. Even though there is no significant difference, if a distinction is needed, a prime mark is used next to the letter. The asterisk indicates statistical significance.

**Table 2 bioengineering-11-00215-t002:** Median with minimum and maximum values of all groups measured by histomorphometric analysis.

Week	Group		TA		NB		%NB		RBG			%RBG	
2 weeks(n = 5)	90m		15.090 (11.450–16.170)	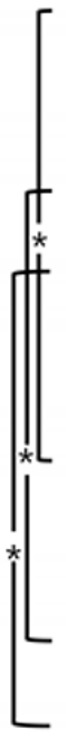	0.960 (0.420–1.240) ^a^	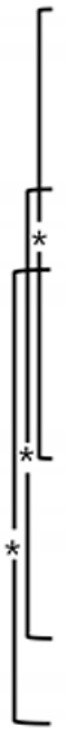	5.937 (3.368–8.217) ^a^	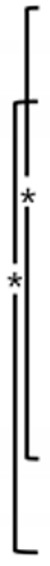	2.150 (1.470–3.990) ^a^		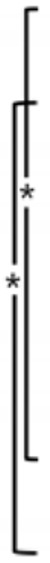	14.515 (12.838–26.441) ^a^	
90n		17.010 (11.240–19.170)	0.200 (0.060–2.840)	1.779 (0.495–15.477)	1.340 (0.920–10.430) ^a′^		11.744 (7.302–54.408) ^a′^	
76m		14.690 (11.680–16.160)	1.160 (0.030–1.470) ^a′^	7.612 (0.257–10.007) ^a′^	4.400 (2.050–6.720)		29.952 (14.012–43.664)	
76n		13.560 (11.780–22.560)	0.990 (0.360–2.460) ^a′′^	6.033 (3.056–10.904) ^a′′^	2.780 (0.400–7.770)		20.501 (3.040–34.441)	
Overall		14.890 (11.240–22.560)	0.975 (0.030–2.840)	5.985 (0.257–15.477)	2.405 (0.400–10.430)		15.300 (3.040–54.408)	
8 weeks(n = 5)	90m	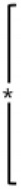	11.755 (10.790–16.770) ^A^	2.245 (1.480–3.920) ^A^	19.840 (12.726–27.625) ^A^	0.155 (0.000–1.390) ^A^	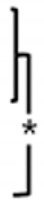	1.196 (0.000–9.796) ^A^	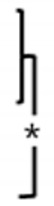
90n	14.735 (9.970–17.130)	2.015 (1.160–3.620)	15.691 (8.033–21.133)	0.180 (0.000–1.010) ^A′^	1.087 (0.000–6.994) ^A′^
76m	13.070 (9.940–16.450)	2.465 (1.310–5.030) ^A′^	16.479 (11.313–50.604) ^A′^		3.580 (0.170–4.580) ^B^		24.323 (1.468–32.068) ^B^
76n	16.720 (13.190–20.550) ^B^	2.505 (1.120–3.320) ^A′′^	15.688 (7.349–16.831) ^A′′^		3.615 (0.000–7.650)			21.968 (0.000–37.226)	
Overall		14.315 (9.940–20.550)		2.290 (1.120–5.030)		16.128 (7.349–50.604)		0.800 (0.000–7.650)			6.034 (0.000–37.226)	

TA: Total augmented area, NB: New bone area, %NB: ratio of NB to TA, RBG: residual bone graft area, %RBG: ratio of RBG to TA. Significant differences (*p* < 0.05) between the four experimental groups are indicated by different small letters at 2 weeks and by different capital letters at 8 weeks. Small and capital letters of the same alphabet indicate significant differences between 2-week and 8-week results compared in the same experimental group. Even though there is no significant difference, if a distinction is needed, a prime mark is used next to the letter. The asterisk indicates statistical significance.

## Data Availability

The datasets generated or analyzed during the current study are available from the corresponding author upon reasonable request.

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
