# Peer review of "Comparative Analysis of Bone Regeneration According to Particle Type and Barrier Membrane for Octacalcium Phosphate Grafted into Rabbit Calvarial Defects"

_bioengineering, 2024, doi:10.3390/bioengineering11030215_

Round 1

Reviewer 1 Report

Comments and Suggestions for Authors

This manuscript evaluates the efficacy of bone formation and volume for different bone substitute materials in a rabbit calvarial defect model. The study is significant because it addresses an important topic related to bone regeneration. However, I think the reported data is not sufficient to create a high impact in the field as there are major shortcomings or missing data in its current form. Positive and negative control groups are missing. The methods are not described in sufficient detail to be reproduced. Bone substitute materials were not characterized sufficiently. Data reporting (tables and histological images) is not done appropriately.

Specific comments:

·         The bone-filling/substitute materials need to be characterized to show the differences. A particle diameter distribution, FTIR as well as a test showing OCP content should be included.

·         Describe clearly how OCP powders were synthesized, suspended, and dried.

·         Describe advanced granulation techniques and provide an image of experimental set-up.

·         Describe the sterilization process with details.

·         Animals: male, female?

·         Provide standards mentioned in the text for the animal study? Or refer to another publication or document.

·         Provide the product number and manufacturer of the collagen membrane.

·         Provide details of animal number determination in statistical analysis section.

·         Provide details of careful clinical observation of the animals throughout the healing period.

·         Positive (native bone) and negative (unoperated defect) control groups are needed.

·         Figure1: a scale bar is needed.

·         Also report the parameters tested for inflammation, allergic reaction, postoperative bleeding, and infection around the surgical site.

·         Provide uCT parameters.

·         Also report Masson trichrome stained sections.

·         Include the stainings for residual particles, connective tissue, adipose tissue, and blood vessels within the defect area.

·         Clearly indicate a significant difference between groups in Tables.

·         Figure3 is not informative. Scale bar shows 1mm while a defect size of 8mm was used. Please include staining images showing the entire defect size.

Author Response

Dear Reviewers

We thank the reviewers for their comments, and we are grateful for the opportunity to provide further revisions to our paper. We changed our manuscript according to the reviewers’ comments and recommendations. We are trying to adequately address each of the points made by the reviewers. We would be very thankful if you could please reconsider a thoroughly revised manuscript. We highlighted the changes made in the manuscript by using a different color font (red): see correction marked form file, and explained details in this letter.

Reviewer 1

Comment:

Specific Comment 1:

         The bone-filling/substitute materials need to be characterized to show the differences. A particle diameter distribution, FTIR as well as a test showing OCP content should be included.

Response 1: 

We have added considerable content in Sections 2.2 and 3.1, which include XRD test results, particle size, and SEM images, hence these can be referred to for further details.

Specific Comment 2:

Describe clearly how OCP powders were synthesized, suspended, and dried.

Response 2: 

The manufacturing process of the OCP powder we provide has already been detailed in Section 2.1.

Specific Comment 3:

Describe advanced granulation techniques and provide an image of experimental set-up.

Response 3: 

Please understand that we are unable to provide detailed descriptions or images of the advanced granulation technology, as it is proprietary to the company.

Specific Comment 4:

Describe the sterilization process with details.

Response 4: 

The sterilization process for bone graft materials is generally similar, so in our study, it is noted in the last sentence of section 2.1 that we applied a range of 25-40 kGy. For several decades, a dose of 25 kGy of gamma radiation has been recommended for the terminal sterilization of medical products, including tissue allografts. While many banks adhere to the 25 kGy standard set by the International Atomic Energy Agency (IAEA), some have opted for higher doses, while others have chosen lower doses.

Specific Comment 5:

Animals: male, female?

Response 5: 

The manuscript already specifies 'male' in the text.

Specific Comment 6:

Provide standards mentioned in the text for the animal study? Or refer to another publication or document.

Response 6: 

The fundamental standards for animal experiments were conducted in accordance with the ARRIVE guidelines, as indicated in Section 2.3. We will also add references to previous rabbit studies conducted using the same method.

Specific Comment 7:

Provide the product number and manufacturer of the collagen membrane.

Response 7: 

Since the collagen membrane is a widely used ECM membrane currently in active clinical use, it seems unnecessary to specify the product number. The product name and manufacturer will be indicated as "Bio-Gide, Geistlich Biomaterials, Wolhusen, Switzerland."

Specific Comment 8:

Provide details of animal number determination in statistical analysis section.

Response 8: 

This experiment utilized ten rabbits. The number of subjects was determined using G*power, referencing previous studies by Son et al. Details have been included in the statistical analysis section.

Specific Comment 9:

Provide details of careful clinical observation of the animals throughout the healing period.

Response 9: 

The observations during the healing period are already presented in the results section 3.1 under "Clinical Findings." We will provide additional details in the results and elaborate on the observation methods in the method section. There were no severe postoperative complications, such as bleeding and swelling. The surgical sites of rabbits healed well without any signs of infection or flap exposure.

Specific Comment 10:

Positive (native bone) and negative (unoperated defect) control groups are needed.

Response 10: 

Comparative studies with control groups from previous research have been completed. References for comparative studies between OCP (synthetic bone) and other natural bones have been added, and it has been noted in Discussion section that positive and negative control groups were not established.

Specific Comment 11:

Figure1: a scale bar is needed.

Response 11: 

We have added a scale bar to Figure 1.

Specific Comment 12:

Also report the parameters tested for inflammation, allergic reaction, postoperative bleeding, and infection around the surgical site.

Response 12: 

There were no indications of inflammation, allergic reactions, postoperative bleeding, or infections around the surgical site.

Specific Comment 13:

Provide uCT parameters.

Response 13: 

We have added the micro-CT settings to Section 2.6.

Specific Comment 14:

Also report Masson trichrome stained sections.

Response 14: 

Masson's trichrome staining was used for measurements for histomophometric analysis. This is well explained in the schematic diagram in figure1.

Specific Comment 15:

Include the stainings for residual particles, connective tissue, adipose tissue, and blood vessels within the defect area.

Response 15: 

Because the part we want to focus on and explain is the formation of new bone around residual particles, we did not add explanations about other tissues.

Specific Comment 16:

Clearly indicate a significant difference between groups in Tables.

Response 16: 

A figure was added to the results in the table, displayed as a bar graph, and statistical significance was written in English letters.

Specific Comment 17:

Figure3 is not informative. Scale bar shows 1mm while a defect size of 8mm was used. Please include staining images showing the entire defect size.

Response 17: 

We have annotated the necessary information in Figure 3 and also specified the defect size.

Reviewer 2 Report

Comments and Suggestions for Authors

Overall the study reported an interesting finding. However, the concerns indicated comments below need to addressed adequately:

1. Why the word "different' is needed? The title can be restructured to add clarity to the title

2. Abstract (line 14)- the synthetic bone is made of OCP? need more clarity in the first 2 sentence of the abstract to understand the aim of the study
3. Line 75-77 better reflect the aim of the study instead of the aims written in the abstract section.

4. Was the OCP synthesized in the authors lab was characterized and any variations were standardized? What is the physicochemical-mechanical properties of the material? What is the difference between the 76% and 90% OCP scaffolds? What is the basis for choosing this 2 % of OCP?

5. In fig 2, the overall images can be labelled to indicate which site correspond which group.
6.  The Table 1  & 2 are better presented in bar charts

7. The results in section 3.3 can be better shown by indicating them in the histological images. 

8. Why no control groups were included? neither no treatment group or comparison with TCP or other gold standard scaffolding was included in this study.

9. From the Fig 2 and BV values , we can see that the 76% OCP enable faster bone regeneration in 2 weeks animal study, but why this is not indicated in conclusion?

10. Lastly, what is the novelty of this study compared to all the previously reported  bone regeneration study in animals using OCP scaffolds?

Comments on the Quality of English Language

Acceptable with minor edits.

Author Response

Reviewer 2

Dear Reviewers

We thank the reviewers for their comments, and we are grateful for the opportunity to provide further revisions to our paper. We changed our manuscript according to the reviewers’ comments and recommendations. We are trying to adequately address each of the points made by the reviewers. We would be very thankful if you could please reconsider a thoroughly revised manuscript. We highlighted the changes made in the manuscript by using a different color font (red): see correction marked form file, and explained details in this letter.

Comment 1:

Why the word "different' is needed? The title can be restructured to add clarity to the title.

Response 1: 

We included the word 'different' in the title to indicate the significant difference in results between the granule and powder types. However, we have restructured the title following your advice.

Comment 2:

Abstract (line 14)- the synthetic bone is made of OCP? need more clarity in the first 2 sentence of the abstract to understand the aim of the study.

Response 2: 

We followed your recommendations. We have added detailed information about OCP in the second sentence of the abstract.

Comment 3:

Line 75-77 better reflect the aim of the study instead of the aims written in the abstract section.

Response 3: 

We have revised the research objectives in the abstract to match those stated in lines 75-77.

Comment 4:

Was the OCP synthesized in the authors lab was characterized and any variations were standardized? What is the physicochemical-mechanical properties of the material? What is the difference between the 76% and 90% OCP scaffolds? What is the basis for choosing this 2 % of OCP?

Response 4: 

The manufacturing method and characteristics of OCP material have been explained in several additional sections. All modifications were carried out using a standardized method, and the particle type was determined and manufactured in consideration of clinical usefulness, and then the concentration of OCP was measured.

Comment 5:

In fig 2, the overall images can be labelled to indicate which site correspond which group.

Response 5: 

We have clearly labeled the groups in the two images corresponding to the overall images in Figure 2.

Comment 6:

The Table 1 & 2 are better presented in bar charts.

Response 6: 

A figure was added to the results in the table, displayed as a bar graph, and statistical significance was written in English letters.

Comment 7:

The results in section 3.3 can be better shown by indicating them in the histological images.

Response 7: 

We have inserted arrows to indicate results in the histological images.

Comment 8:

Why no control groups were included? neither no treatment group or comparison with TCP or other gold standard scaffolding was included in this study.

Response 8: 

We have added references for the aspects of this study that have already been compared in previous research.

Comment 9:

From the Fig 2 and BV values , we can see that the 76% OCP enable faster bone regeneration in 2 weeks animal study, but why this is not indicated in conclusion?

Response 9: 

While the two-week results for 76% OCP showed a significantly larger bone volume, it was difficult to confirm a substantial new bone formation, hence it was not indicated. Following your advice, we will conclude that the bone volume was maintained more significantly.

Comment 10:

Lastly, what is the novelty of this study compared to all the previously reported bone regeneration study in animals using OCP scaffolds?

Response 10: 

In comparison to previous studies using OCP, this study holds significance in directly comparing the results based on the application type of OCP bone substitute and the use of collagen membrane, thereby confirming its clinically meaningful application.

Reviewer 3 Report

Comments and Suggestions for Authors

The reviewer appreciates the efforts of the authors to conduct this study with good clinical significance. The study is well-designed to achieve the objectives. The manuscript is well written however, the reviewer noticed a few errors that need to be revised before acceptance.

The information related to materials used in the study is missing. The reviewer recommends adding a table with the list of materials and product details (manufacturer, county, composition batch/ lot number) wherever applicable. Alternatively, the author can incorporate them as a text in the materials and methods section. 

Is the study design (calvarial defects in rabbits) following any previously published article? If so then add the citation for the same. If it is a new study design then the author should add rationale and justification for the selected animal model

Tables 1 and 2 resolutions are poor. Please follow the journal instructions for the tables

Please add scale in micro-CT images

Please highlight the significance in Figure 3 by adding an arrow/ circle/ any symbol for easy understanding by the reader.  

The reviewer highly recommends adding a graphical diagram/ flowchart to explain the methodology sequence and group distribution

Author Response

Reviewer 3

We thank the reviewers for their comments, and we are grateful for the opportunity to provide further revisions to our paper. We changed our manuscript according to the reviewers’ comments and recommendations. We are trying to adequately address each of the points made by the reviewers. We would be very thankful if you could please reconsider a thoroughly revised manuscript. We highlighted the changes made in the manuscript by using a different color font (red): see correction marked form file, and explained details in this letter.

Comment 1:

The information related to materials used in the study is missing. The reviewer recommends adding a table with the list of materials and product details (manufacturer, county, composition batch/ lot number) wherever applicable. Alternatively, the author can incorporate them as a text in the materials and methods section.

Response 1: 

I have added detailed information about the materials used in the study using the text from the Materials and Methods section.

Comment 2:

Is the study design (calvarial defects in rabbits) following any previously published article? If so then add the citation for the same. If it is a new study design then the author should add rationale and justification for the selected animal model.

Response 2: 

We have added references as this study follows the model of a previous research.

Comment 3:

Tables 1 and 2 resolutions are poor. Please follow the journal instructions for the tables.

Response 3: 

We have reproduced Tables 1 and 2 following the journal guidelines for tables.

Comment 4:

Please add scale in micro-CT images.

Response 4: 

We have added scale bars to the micro-CT images.

Comment 5:

Please highlight the significance in Figure 3 by adding an arrow/ circle/ any symbol for easy understanding by the reader.

Response 5: 

We have added arrows and circles to the histological results figure in Figure 3 to indicate newly formed bone tissue and features.

Comment 6:

The reviewer highly recommends adding a graphical diagram/ flowchart to explain the methodology sequence and group distribution

Response 6: 

We have added a diagram to illustrate the flow of the research.

Round 2

Reviewer 1 Report

Comments and Suggestions for Authors

This manuscript evaluates the efficacy of bone formation and volume for different bone substitute materials in a rabbit calvarial defect model. The study is significant because it addresses an important topic related to bone regeneration. However, I think the reported data still needs to be improved. The methods are not described in sufficient detail to be reproduced. Data reporting (tables and histological images) is not done appropriately.

Specific comments:

·         A particle diameter distribution is needed in Fig3, this is important because in one group the authors are using powders and in the other one granules. In Fig 4, the structure looks similar. Also, a scale bar is missing in Fig3.

·         Describe clearly how OCP powders were synthesized, suspended, and dried. I believe the procedure is still not sufficiently detailed to reproduce.

·         Describe advanced granulation techniques and provide an image of experimental set-up. This is needed for the clarity of the procedure. A patent can be referred to if available.

·         The sterilization process needs to be detailed further. Only 25-40kGy was included. The process should be detailed enough to be reproduced by others.

·         Provide the product number and manufacturer of the collagen membrane. The company offers other products in the market. A product number is needed.

·         In Table 1&2: It is not clear what the English letters are referring to. Please specify below the table as done for TV, BV, etc..

Author Response

This manuscript evaluates the efficacy of bone formation and volume for different bone substitute materials in a rabbit calvarial defect model. The study is significant because it addresses an important topic related to bone regeneration. However, I think the reported data still needs to be improved. The methods are not described in sufficient detail to be reproduced. Data reporting (tables and histological images) is not done appropriately.

Specific comments:

1) A particle diameter distribution is needed in Fig3, this is important because in one group the authors are using powders and in the other one granules. In Fig 4, the structure looks similar. Also, a scale bar is missing in Fig3.

Answer) The authors deeply appreciate the reviewer's meticulous comments. We thoroughly agree that the aspect related to particle size is among the most crucial elements in this study and desire to ensure its reflection in the text. Through several revisions, we have endeavored to incorporate related content into the manuscript as comprehensively as possible and have described the characteristics of granules and powders using an X-ray diffractometer (XRD).

Figure 3A demonstrates that both granules and powders, irrespective of particle size, are primarily composed of OCP without any residual DCPD.

By adding lines 105-109, we clarified that the material comprised two types of crystals with non-uniform sizes from the time of manufacture. The first type of crystal was recorded to have a length of 10-15 µm and a width of 2-3 µm, while the second type had a length of 1-10 µm and a width of 0.5-3 µm.

Although the structure may appear similar in Figure 4, observations from an electron microscope in Figure 3B showed a wide range of sizes from a few micrometers to several tens of micrometers in both granules and powders. Therefore, due to the broad and diverse particle size distribution of the OCP used in this study, defining a single range for all materials used in every specimen could be challenging and potentially misrepresentative. Given that there have been previous studies designing experimental groups based on particle size, referencing this paper could differentiate our study from others (Appl. Sci. 2021, 11(17), 7921; https://doi.org/10.3390/app11177921).

The authors focused on creating clinically manageable forms of the OCP material rather than on particle size and aimed to assess how the application method of granules or powders might affect new bone formation. For this purpose, the use of a barrier membrane was also included as part of this study's experimental groups, similarly to other GBR materials, and significant differences in the results of new bone formation were observed depending on the form of the graft and the presence of a membrane.

Furthermore, following the reviewer's advice, a scale bar has been added to Figure 3. We extend our gratitude once again for your detailed review.

2) Describe clearly how OCP powders were synthesized, suspended, and dried. I believe the procedure is still not sufficiently detailed to reproduce.

Answer) In lines 88-109, we have provided detailed descriptions of the synthesis process for OCP powder, highlighted with red underlining in the text.

3) Describe advanced granulation techniques and provide an image of experimental set-up. This is needed for the clarity of the procedure. A patent can be referred to if available.

Answer) Following the reviewer's recommendation, we have detailed the granulation technique in lines 111-118, referencing the patent.

4) The sterilization process needs to be detailed further. Only 25-40kGy was included. The process should be detailed enough to be reproduced by others.

Answer) In response to the reviewer's comments, we have elaborated on the sterilization process in lines 122-128.

5) Provide the product number and manufacturer of the collagen membrane. The company offers other products in the market. A product number is needed.

Answer) Following the reviewer's comments, we have added the product number for the collagen membrane in lines 153.

6)  In Table 1&2: It is not clear what the English letters are referring to. Please specify below the table as done for TV, BV, etc.

Answer) In our previous revisions, we have already provided explanations for the abbreviations in English at the bottom of Tables 1 and 2.

Reviewer 2 Report

Comments and Suggestions for Authors

The revision made satisfactorily. The reply given for the comment no.9 & 10 should be included into the main text.  

Author Response

The revision made satisfactorily. The reply given for the comment no.9 & 10 should be included into the main text.

Answer) All authors deeply appreciate the substantial improvements made to the submitted manuscript, thanks to the meticulous comments provided by the reviewer. In response to items 9 and 10 mentioned by the reviewer in Round 1, we have added the text to the Discussion section (Line 373-375 & 409-412)

Round 3

Reviewer 1 Report

Comments and Suggestions for Authors

This manuscript evaluates the efficacy of bone formation and volume for different bone substitute materials in a rabbit calvarial defect model. The study is significant because it addresses an important topic related to bone regeneration. Minor comments need to be addressed before the manuscript can be accepted.

Specific comments:

·         Provide an image of the experimental set-up for granulation techniques. “….. we have detailed the granulation technique in lines 111-118, referencing the patent.” No reference was provided.  

·         In Table 1&2: It is unclear what the English letters refer to. In the previous or revised version, such explanation is not seen. Please specify below the table as done for TV, BV, etc..

Author Response

<Reviewer Comment>

This manuscript evaluates the efficacy of bone formation and volume for different bone substitute materials in a rabbit calvarial defect model. The study is significant because it addresses an important topic related to bone regeneration. Minor comments need to be addressed before the manuscript can be accepted.

Specific comments:

  • Provide an image of the experimental set-up for granulation techniques. “….. we have detailed the granulation technique in lines 111-118, referencing the patent.” No reference was provided.

Answer) We followed the reviewer's recommendation and introduced key points about the OCP particle formation process in lines 111-115 and provided the reference below. This process is discussed in detail in the <preparation of OCP composite> section of the literature, so we are confident that sufficient information will be provided.

Jeong, C.H.; Kim, J.; Kim, H.S.; Lim, S.Y.; Han, D.; Huser, A.J.; Lee, S.B.; Gim, Y.; Ji, J.H.; Kim, D.; et al. Acceleration of bone formation by octacalcium phosphate composite in a rat tibia critical-sized defect. J. Orthop. Transl. 2022, 37, 100–112.

  • In Table 1&2: It is unclear what the English letters refer to. In the previous or revised version, such explanation is not seen. Please specify below the table as done for TV, BV, etc.

Answer) Based on the reviewer's comments, we added explanations of abbreviations and English letters at the bottom of Tables 1 and 2 as follows.

Significant differences (p<0.05) between the four experimental groups are indicated by different small letters at 2 weeks and by different capital letters at 8 weeks. Small and capital letters of the same alphabet indicate significant differences between 2-week and 8-week results compared in the same experimental group. Even though there is no significant difference, if a distinction is needed, a prime mark is used next to the letter.
